# *Arctoscopus japonicus* Lipids Enhance Immunity of Mice with Cyclophosphamide-Induced Immunosuppression

**DOI:** 10.3390/foods12173292

**Published:** 2023-09-01

**Authors:** JeongUn Choi, Weerawan Rod-in, A-yeong Jang, Woo Jung Park

**Affiliations:** 1Department of Wellness-Bio Industry, Gangneung-Wonju National University, Gangneung 25457, Republic of Korea; 3je0ngun2@gmail.com; 2Department of Marine Food Science and Technology, Gangneung-Wonju National University, Gangneung 25457, Republic of Korea; weerawan.ve@gmail.com (W.R.-i.); jay941006@gmail.com (A.-y.J.)

**Keywords:** lipids, immunomodulatory, cyclophosphamide, immunosuppression

## Abstract

A lipid extract was obtained from eggs of the sailfin sandfish, *Arctoscopus japonicus*. Immunostimulatory effects of *A. japonicus* lipids incorporated with PEG6000 (AJ-PEG) on immunosuppressed mice treated with cyclophosphamide (CY) were investigated. AJ-PEG was administered orally to mice at different concentrations of 25 to 100 mg/kg body weight (BW). CY was injected to mice intraperitoneally at 80 mg/kg BW. Administration of AJ-PEG significantly increased the spleen index of CY-treated mice. AJ-PEG also stimulated the proliferation of splenic lymphocytes and natural killer (NK) activity. Immune-associated cytokines such as *IL-1β*, *IL-2*, *IL-4*, *IL-6*, *TNF-α*, and *IFN-γ* as well as *TLR4* were overexpressed in splenic lymphocytes. Furthermore, AJ-PEG significantly increased splenic CD4+ and CD8+ T lymphocytes. In peritoneal macrophages, AJ-PEG administration improved proliferation, nitric oxide (NO) production, and phagocytosis. It also upregulated *iNOS*, *COX-2*, *IL-1β*, *IL-6*, and *TNF-α* expression. Taken together, these results suggest that AJ-PEG can be used in animal models with immunosuppressive conditions as a potent immunomodulatory agent.

## 1. Introduction

The immune system consists of cells and chemicals that play specific roles in defending against infection. It has two categories: innate immunity and adaptive immunity. The immune system protects the body from microorganisms and foreign antigens [1]. Several diseases, such as autoimmune diseases, inflammatory diseases, and even cancer, are caused by problems with the immune system [2]. Cyclophosphamide (CY) is a chemotherapy drug commonly used for malignant tumors and auto-immune diseases. However, it has a number of serious side effects, including nausea, fatigue, and immunosuppression [3,4]. CY-induced immunosuppressed mice have been used in many studies to evaluate immunomodulatory effects of functional materials [4,5,6]. These studies have assessed immune organ indices (spleen and thymus) and immune functions by measuring the activation of immune-related cells such as macrophages, lymphocytes, and natural killer (NK) cells that can secrete cytokines and chemokines for lymphocyte proliferation [6,7,8,9]. Macrophages are phagocytic cells generated from monocytes that play important roles in both adaptive and innate immunity [4]. In inflammation, functional responses of macrophages are involved in protecting the host by phagocytosis, antigen presentation, and immunomodulation that release various cytokines and growth factors [10].

Lipids and fatty acids are the main structural components of cell membranes. They are essential for several metabolic processes and some cellular functions [11]. Marine fish eggs contain high levels of polyunsaturated fatty acids (PUFAs) such as eicosapentaenoic acid (EPA, 20:5n3) and docosahexaenoic acid (DHA, 22:6n3), which provide energy for fish to survive and reproduce [12].

Fatty acids are also used as building blocks for the biosynthesis of membranes as intracellular signaling molecules and precursors for the production of other cellular components [11]. Fatty acids might be indirectly effective for many inflammatory symptoms and diseases in which they have been shown to reduce inflammatory responses in patients with Alzheimer’s disease, cancer, cardiovascular disease, and anti-inflammatory conditions [13,14,15,16], with many clinical references [17,18,19,20]. In immune cells, fatty acids are responsible for generating energy, regulating gene expression, and producing bioactive lipid mediators [21,22]. Recently, DHA has been shown to be able to improve immunomodulatory activity and molecular mechanism in RAW264.7 cells by stimulating GPR120 and MAPKs to NF-κB pathways [23,24]. EPA and DHA-enriched fish oil has beneficial effect on the immune system of patients with breast cancer [25]. Supplementation of fish oil in early infancy has an effect on the development of infant immune responses [26]. A suitable dietary DHA/EPA ratio not only improves the growth performance of marine fish, but also improves their immune responses [27]. In addition, monounsaturated fatty acids (MUFAs), especially oleic acid (OA, 18:1n9), can modulate inflammation and boost reparative activity in wounds [28,29]. Saturated fatty acids (SFAs), such as palmitic acid (16:0) and stearic acid (18:0), are also involved in immune response [30,31].

The sailfin sandfish (*Arctoscopus japonicus*) is a cold-water species belonging to the family Trichodontidae. It has wide range of distribution, including the North Western Pacific Ocean, east coast of Korea, North central Japan, Sakhalin, and Alaska [32,33]. *A. japonicus* is a popular commercial fish species in the Republic of Korea [34] due to its texture, taste, and health benefits [35]. It has been reported that the meat and eggs of *A. japonicus* contain functional peptides with antioxidant and anti-inflammatory properties [35,36,37,38]. A peptide derived from enzymatic hydrolysates can inhibit inflammation in RAW264.7 cells [37,38]. The main fatty acids found in Japanese sandfish eggs were palmitic acid, oleic acid, DHA, and EPA [39]. Our previous studies have shown that lipids isolated from *A. japonicus* eggs, which contain high levels of PUFAs, especially EPA and DHA, possess immune biological activities in mouse RAW264.7 macrophage cells [40,41]. However, immune-enhancing activities of *A. japonicus* lipids in *in vivo* immunosuppressive mouse systems have not yet been reported. Therefore, the current study evaluated effects of oral administration of lipids extracted from *A. japonicus* eggs on mice with immunosuppression induced by CY.

## 2. Materials and Methods

### 2.1. Animals

Six-week-old BALB/c mice with a mean weight of 21–23 g were obtained from Central Lab Animal Inc. (Seocho, Republic of Korea). All mice were reared at 22 ± 2 °C with 55–60% humidity and a 12-h light–dark cycle. They had free access to water and food throughout the experiment. The Institutional Animal Care and Use Committee (IACUC) of Gangneung-Wonju National University committee approved this experiment (approval number: GWNU-2021-12).

### 2.2. Isolation of Lipids from A. japonicus Eggs

*A. japonicus* was obtained from the East Sea in Gangwon, Republic of Korea. *A. japonicus* eggs were separated, freeze-dried, and crushed for lipid extraction. *A. japonicus* lipids (AJ) were prepared by extraction using a modified Bligh and Dyer’s method [42]. Gas chromatography (GC)-flame ionization detection (FID) analysis was used to determine fatty acid profiles of AJ. Extracted lipids with a yield of 2.18 g (*w*/*w*) or 6.92% of dry material were dissolved in absolute ethanol.

### 2.3. Preparation of AJ-PEG

AJ-PEG was prepared from AJ and PEG6000 as previously described [43]. Normally, saline is used for oral administration. However, saline could not dissolve our sample of AJ. Therefore, PEG6000 was used because of its hydrophilic properties in covering hydrophobic molecules of lipids, and then AJ-PEG was dissolved in saline. Mixtures of AJ and PEG6000 (1:1, *w*/*w*) were melted in a water bath at 60 °C. A rotary evaporator was used to remove the solvent at 40 °C for 2 h at 45 rpm. Dried samples were stored at −20 °C for another 24 h. AJ-PEG was stored in a desiccator after samples were crushed and sieved through a 100-mesh.

### 2.4. Establishment of Immunosuppressed Mice Models

After environmental adaption, mice were randomly divided into nine groups (five mice in each group) as follows: normal group, CY group, ginseng group, levamisole group, PEG6000 group, and various concentrations of AJ-PEG (Table 1). Both levamisole and ginseng were used as positive controls. PEG6000 group was used as a model control group. From the 1st day to the 10th day, daily gavage was performed for mice through oral administration. All mice (except for the normal group) were administered by intraperitoneal (i.p.) injection with CY (80 mg/kg BW) from the 4th day to the 6th day. Mice were sacrificed 24 h after the final dose.

### 2.5. Preparation of Peritoneal Macrophages

The Ray method [44] was used to collect peritoneal macrophages. Cells were harvested by peritoneal lavage with 5 mL of ice-cold phosphate buffered saline (PBS) and 3% fetal bovine serum (FBS). Cell suspension was centrifuged and washed twice with 1× PBS buffer. Peritoneal macrophages were suspended in RPMI-1640 medium supplemented with 10% FBS and 1% penicillin/streptomycin (PS) at a density of 1 × 10^6^ cells/mL.

### 2.6. Isolation of Spleen Lymphocytes

To extract splenocytes, BALB/c mouse spleens were gently disrupted. Based on spleen weight (mg) and body weight (g), spleen index was calculated. After weighing, spleen was isolated using 1 × RBC Lysis Buffer (eBioscience, San Diego, CA, USA). It was then centrifuged and washed with 1 × PBS buffer. Cells were resuspended in RPMI-1640 medium supplemented with 1% FBS and 1% PS at a density of 2 × 10^6^ cells/mL.

### 2.7. Determination of Peritoneal Macrophage Proliferation and Nitric Oxide (NO) Production

Peritoneal macrophages were cultured with or without lipopolysaccharides (LPS, 1 µg/mL). After 24 h of incubation, culture supernatants (100 µL) were mixed with Griess reagent (Sigma-Aldrich, Saint Louis, MO, USA) and incubated at room temperature for 10 min. The absorbance at 540 nm was measured to determine whether macrophage culture supernatants contained nitrite. Proliferation of cells was determined using an EZ-Cytox Cell Viability Assay Kit (Daeil Lab Service, Seoul, Republic of Korea). To each well, 100 µL of water-soluble tetrazolium (WST) solution was added. After incubation at 37 °C for 1 h, absorbance at 450 nm was measured with a microplate reader (BioTek Instruments, Santa Clara, CA, USA).

### 2.8. Measurement of Peritoneal Macrophage Phagocytosis

Phagocytosis was determined with a neutral red uptake assay as reported previously [45]. Briefly, a neutral red solution (0.09% mass fraction of solute) was added to each well. After incubating at 37 °C for 30 min, cells were rinsed with 1 × PBS buffer to remove excess color. Then 100 μL of 50% EtOH with 1% glacial acetic acid was added. Absorbance at 540 nm was then measured with a microplate reader (BioTek Instruments, USA).

### 2.9. Proliferation of Splenic Lymphocytes

Splenocytes (2 × 10^6^ cells/mL) were stimulated with concanavalin A (Con A, 5 µg/mL) as T cell mitogen, and LPS (10 µg/mL) as B cell mitogen and cultured at 37 °C for 48 h in a humidified incubator containing 5% CO_2_. Cell proliferation was assessed using the EZ-Cytox Cell Viability Assay Kit (DaeilLab Service, Seoul, Republic of Korea).

### 2.10. NK Cell Activity of Splenocytes

Splenocytes were obtained and treated with YAC-1 cells at a ratio of effector to target cells of 50:1. After incubation at 37 °C for 24 h, cells were used for CytoTox 96^®^ Non-Radioactive cytotoxicity assay (Promega, Madison, WI, USA) according to the manufacturer’s instructions.

### 2.11. Measurement of Immune-Associated Gene Expression

Total RNAs were isolated from cells using a Universal RNA Extraction Kit (Takara Bio Inc., Tokyo, Japan) according to the manufacturer’s instructions. A high capacity cDNA reverse transcription kit (Applied Biosystems, Saint Louis, MO, USA) was used to prepare cDNAs. Immune-associated gene expression was determined using a QuantStudio™ 3 FlexReal-Time PCR System (Applied Biosystems, USA). PCR reaction consisted of 5 ng/µL of cDNA mixed with TB Green^®^ Premix Ex Taq™ II (Takara Bio Inc., Tokyo, Japan) and specific primers (Table 2).

### 2.12. Measurement of T-Lymphocyte Subsets in the Spleen

A splenocyte suspension was incubated at 4 °C for 20 min with 0.5 µL of anti-CD3e-PE, 1.5 µL of anti-CD4-APC, and 1 µL of anti-CD8-FITC antibodies. Cells were centrifuged and washed twice with 1 × PBS buffer before being resuspended in FACs buffer (2% FBS and 0.1% sodium azide in 1 × PBS buffer). A CytoFLEX Flow Cytometer (Beckman Coulter, Inc., Brea, CA, USA) was used to count CD4+ and CD8+ T lymphocytes.

### 2.13. Statistical Analysis

All data analyses were performed with SPSS 23.0 software (SPSS, Armonk, NY, USA). Data were compared using one-way ANOVA and Duncan’s multiple-range test with a significance level set at *p* < 0.05. Values are presented as means ± standard deviation (SD).

## 3. Results

### 3.1. Fatty Acid Profiles of A. japonicus Lipids Incorporated with PEG6000 (AJ-PEG)

In general, there are three main types of fatty acids present in marine fish eggs: SFAs, MUFAs, and PUFAs [12,39]. As shown in Figure 1, GC-FID results indicated that AJ-PEG was composed of SFAs (28.3 ± 0.5%), MUFAs (29.6 ± 0.3%), and PUFAs (42.1 ± 0.9%). The most predominant compounds were 16:0 (24.4%), DHA (20.6%), 18:1n9 (19.9%), EPA (15.9%), 18:1n7 (6.7%), and 18:0 (4.0%). Similarly, several studies have shown that these fatty acids can modulate immune activity [22,23,24,29,46].

### 3.2. Effect of AJ-PEG on Spleen Index

Based on immune organ indices, the spleen and thymus play an important role in innate immunity [2]. Figure 2 shows spleen size and spleen index of mice with CY-induced immunosuppression. Spleen size (Figure 2A) and spleen index (Figure 2B) of the CY group were decreased significantly compared to those of the normal group. The spleen index of AJ-PEG treated group was significantly increased compared to that of CY treated group, suggesting that AJ-PEG could increase the size and index of the spleen of immunosuppressed mice caused by CY. However, PEG6000, 25 mg/kg BW AJ-PEG, and CY groups showed no statistically significant differences in spleen size or spleen index. Additionally, the spleen index was significantly increased in levamisole and ginseng groups in comparison with that in the CY group.

### 3.3. Effect of AJ-PEG on Splenic Lymphocyte Proliferation

The CY group showed a significant reduction in the proliferation of splenic lymphocytes in response to both T-and B-cell mitogens (Figure 2C). Combining with Con A or LPS, AJ-PEG enhanced the proliferation of splenic lymphocytes of immunosuppressed mice. A dose-dependent increase in splenocyte proliferation was observed in the group treated with AJ-PEG at 25, 50, 75, or 100 mg/kg BW compared with that of the CY group. In addition, both Con A and LPS-stimulated groups in the 75 mg/kg BW of AJ-PEG showed similar results to the normal group. Thus, AJ-PEG could enhance Con A- and LPS-induced proliferation of splenic lymphocytes in CY-treated mice.

### 3.4. Effect of AJ-PEG on NK Cell Activity of Splenocytes

Splenocytes were evaluated for their cytotoxic activity against NK-sensitive YAC-1 cells. As shown in Figure 2D, CY suppressed splenic NK cell activity compared with the normal group. In CY-induced mice, the groups of PEG6000, ginseng, levamisole, and AJ-PEG (25–100 mg/kg BW) gradually stimulated the NK cell activity. AJ-PEG increased NK cell activity in a dose-dependent manner. Specifically, AJ-PEG at 75 mg/kg BW restored cytotoxic activity of CY-treated group to a level similar to that of the normal group.

### 3.5. Effect of AJ-PEG on Gene Expression in Splenic Lymphocytes

As shown in Figure 3, CY group had lower expression levels of immune-associated genes in spleen lymphocytes compared to the normal group. Treatment with CY remarkably down-regulated the expression of immune-associated genes in response to T- and B-cell mitogens. Expression levels of immune-associated genes such as interleukin (*IL*)*-1β*, *IL-2*, *IL-4*, *IL-6*, tumor necrosis factor (*TNF*)-*α*, interferon (*IFN*)-*γ*, and toll-like receptors 4 (*TLR4*) in splenocytes of groups treated with AJ-PEG at different concentrations (25–100 mg/kg BW) were enhanced significantly compared to those in the CY group. The group treated with AJ-PEG at 100 mg/kg BW exhibited lower expression levels than the group treated with AJ-PEG at 75 mg/kg BW. Especially, cells stimulated with Con A were significantly more likely to show high expression levels of these genes than those stimulated with LPS. These results suggest that AJ-PEG could restore immunosuppression of mice by activating Th1 and Th2 cytokines.

### 3.6. Effect of AJ-PEG on T Lymphocytes’ Subsets of Splenocytes

The population was analyzed for expression of CD4+ and CD8+ T lymphocytes by selecting CD3+ for analysis of T cells using flow cytometry. Percentages of CD4+ and CD8+ T lymphocytes (Figure 4A) and the rate of CD4+/CD8+ (Figure 4B) were significantly lower in the CY group. AJ-PEG (75 and 100 mg/kg BW) and positive control groups (ginseng and levamisole) significantly increased the CD4+/CD8+ ratio compared to CY treated group.

### 3.7. Effect of AJ-PEG on Peritoneal Macrophage Proliferation and NO Production

In order to investigate the efficiency of AJ-PEG, peritoneal macrophages from each group of mice were activated with LPS. A dose-dependent improvement in macrophage proliferation and NO production was observed in AJ-PEG-treated mice compared to the CY group (Figure 5A,B). Furthermore, AJ-PEG at 75 and 100 mg/kg BW significantly increased the proliferation of peritoneal macrophages and NO production at a similar level to the normal group.

### 3.8. Effect of AJ-PEG on Peritoneal Macrophage Phagocytosis

As shown in Figure 5C, the peritoneal macrophage phagocytosis of the CY group was significantly lower than that of the normal group. The rate of phagocytosis in normal mouse peritoneal macrophages was considered to be 100% by absorbing neutral red. Compared to the CY group, rates of peritoneal macrophage phagocytosis were markedly increased by ginseng, levamisole, PEG6000, and AJ-PEG (25–100 mg/kg BW). AJ-PEG at 75 mg/kg BW promoted more recovery of macrophage phagocytosis ratio than normal control.

### 3.9. Effect of AJ-PEG on Gene Expression in Peritoneal Macrophages

To further understand the effect of AJ-PEG on peritoneal macrophage activation in CY-treated mice, real-time qPCR was used to determine expression levels of immune-associated genes such as inducible nitric oxide synthase (*iNOS*), cyclooxygenase-2 (*COX-2*), *IL-1β*, *IL-6*, and *TNF-α.* As shown in Figure 6, the mRNA expression levels of these genes were remarkably down-regulated in the CY group. Administration of AJ-PEG (25, 50, 75, and 100 mg/kg BW) significantly increased the expression of several immune-associated genes compared to the CY group. However, AJ-PEG at 100 mg/kg BW showed slightly reduced expression levels compared to other AJ-PEG groups. Additionally, AJ-PEG at 75 mg/kg BW of peritoneal macrophages showed similar or higher gene expression than the positive control group.

## 4. Discussion

Lipids isolated from *A. japonicus* eggs could activate immune function on macrophages in an in vitro cellular system [40]. In the present study, immunomodulatory effects of AJ-PEG on CY-induced immunosuppression were studied in BALB/c mice as an in vivo system. CY-induced immunosuppressed mouse models have been extensively used for evaluating immunomodulatory effects of functional materials [4,5,6]. Furthermore, levamisole and ginseng are known to possess immune regulating and stimulating activities [47,48]. Thus, they were used as positive controls. They are known to activate immune cells, including monocytes/macrophages, lymphocytes, and NK cells, in response to stimuli by performing immune functions [2]. PEG is a synthetic linear polymer that is water-soluble and nonionic with a low toxicity. It has been used in a variety of products as a potent carrier for drug delivery to various organs [49,50]. In addition, polymer-modified recombinant vectors can significantly induce reduced innate immune responses in C57BL/6 mice that regulate the expression of IL-2, IFN-γ, IL-4, and IL-10 [51].

Lymphocytes are major cellular components of the adaptive immune response. Non-specific immunomodulation occurs when T or B cells are stimulated by antigens or mitogens [5]. Cellular proliferation caused by Con A and LPS was used to detect T and B cells in splenocytes [2,5,8]. Splenic NK cells are large granular lymphocytes. When activated by target cells’ recognition and signal integration from both activating and inhibitory receptors, splenic NK cells can directly kill pathogen-infected cells [2]. Our results showed that AJ-PEG treatment could enhance Con A- and LPS-induced splenic lymphocyte proliferation and the NK cell activity of splenic lymphocytes in CY-treated mice, consistent with previous studies on toxicity induced by CY [2,5]. In a similar manner to AJ-PEG, polysaccharides of *Hordeum vulgare* have been reported to have lower splenic NK cytotoxicity at high doses [5]. Glycosaminoglycan of *Apostichopus japonicus* could increase the cytotoxicity of NK cells to improve immunity [6]. These results suggest that AJ-PEG might be able to prevent CY-induced suppression of NK cell activity and lymphocyte proliferation. In addition, T lymphocytes can be divided into different subsets. Total number of T lymphocytes is reflected in the number of CD3+ T cells, while CD4+ and CD8+ including helper and cytotoxic T cells are important effectors of immune cells [52]. Several studies have demonstrated that immunosuppressed mice induced by CY express CD4+ and CD8+ molecules that are indicative of cellular immunity [6,9,53]. The ratio of CD4+/CD8+ cells treated with AJ-PEG increased significantly compared with the negative control group, indicating that AJ-PEG can reverse CY-induced immunosuppression.

Macrophages are highly versatile cells that play a central role in defending their hosts from bacterial infections by nature of their phagocytic, cytotoxicity, intracellular killing ability [54]. NO synthesized by activated macrophages is a major effector molecule in biological functions of innate immune cells [6]. Phagocytosis is a process mediated by a specialized group of innate immune cells called phagocytes, such as macrophages in immune response [2]. Natural substances of various marine sources can stimulate peritoneal macrophages by inducing NO, phagocytosis, and cytokine production to enhance the immune system [6,55]. These results confirmed the non-cytotoxicity of the AJ-PEG and the release of high amounts of NO on peritoneal macrophages in CY-treated mice. Polysaccharides from *Strongylocentrotus nudus* eggs could increase macrophage phagocytosis in the immune system [54]. According to our results, AJ-PEG promoted phagocytosis of peritoneal macrophages, suggesting that AJ-PEG might enhance a nonspecific immune function in mice treated with CY.

Several immune responses are influenced directly and indirectly by cytokines released by Th1 and Th2 cells. Inflammatory cytokines from Th1 cells secreted IL-1, IL-2, IL-12, and IFN-γ, which exhibited cytotoxicity and played a role in cellular immunity. Th2 cells can induce cytokines such as IL-4, IL-6, and IL-10 [56]. Results of the present study showed that AJ-PEG dose-dependently increased splenocyte cytokine (*IL-1β*, *IL-2*, *IL-6*, *TNF-α*, *IFN-γ*, and *TLR4*) expression. A high dose of AJ-PEG (75 mg/kg BW) could stimulate the release of more cytokines. Furthermore, activated macrophages can release a variety of bioactive substances such as several cytokines (IL-1β, IL-2, IL-6, IL-18, and TNF-α), chemokines, inflammatory factors, and reactive oxygen or nitrogen species [6,10]. Peritoneal macrophages also expressed cytokines (*iNOS*, *COX-2*, *IL-1β*, *IL-6*, and *TNF-α*) significantly after treatment with AJ-PEG, similar to splenic cytokine expression. These results suggest that AJ-PEG can enhance the function of splenocyte and peritoneal macrophage in mice treated with CY.

## 5. Conclusions

This study showed that AJ-PEG enhanced the spleen index, splenic lymphocyte proliferation, NK cell activity, and splenic lymphocyte gene expression (*IL-1β*, *IL-2*, *IL-4*, *IL-6*, *TNF-α, IFN-γ*, and *TLR4*), as well as the expression of T lymphocyte subsets (CD4+ and CD8+). Furthermore, AJ-PEG increased cell proliferation, NO production, and phagocytosis by peritoneal macrophages and enhanced the expression of immune-regulated genes (*iNOS*, *COX-2*, *IL-1β*, *IL-6*, and *TNF-α*). These results demonstrated that lipids extracted from *A. japonicus* eggs can improve immune responses in CY-treated immunosuppressed mice.

## Figures and Tables

**Figure 1 foods-12-03292-f001:**
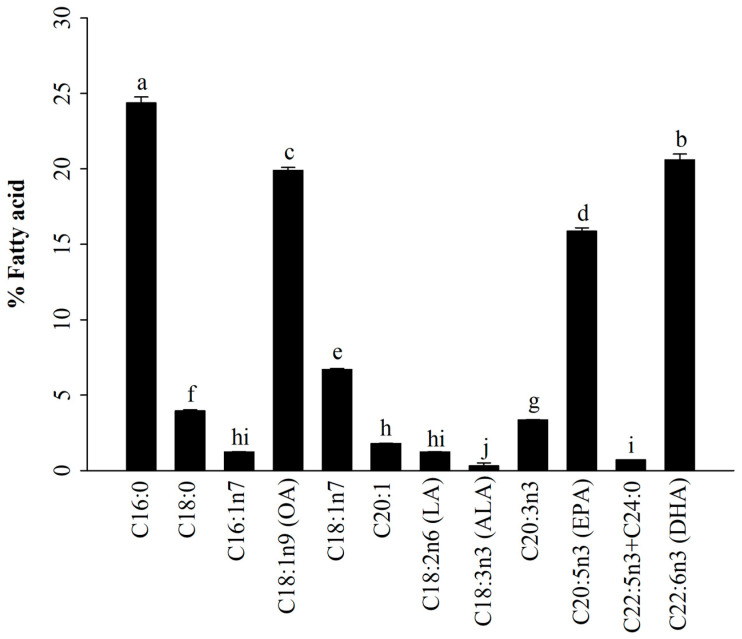
Fatty acid composition of AJ-PEG. Values are presented as means ± SD. Different letters indicate a statistically significant difference at *p* < 0.05 within amounts of total fatty acid from AJ-PEG.

**Figure 2 foods-12-03292-f002:**
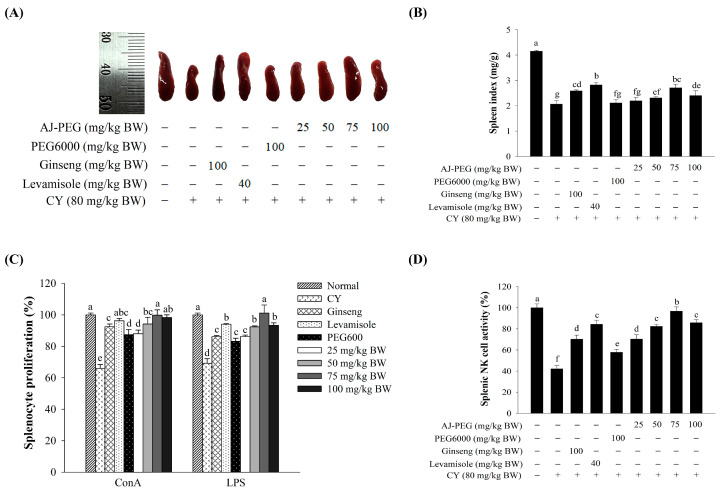
Effects of AJ-PEG at various concentrations on splenocytes. (**A**) Effects on spleen size. (**B**) Effects on spleen index. (**C**) Effects on splenic lymphocyte proliferation. (**D**) Effects on splenic NK cell cytotoxic activity. Values are presented as mean ± SD. Different letters indicate a statistically significant difference at *p* < 0.05 within treatment groups.

**Figure 3 foods-12-03292-f003:**
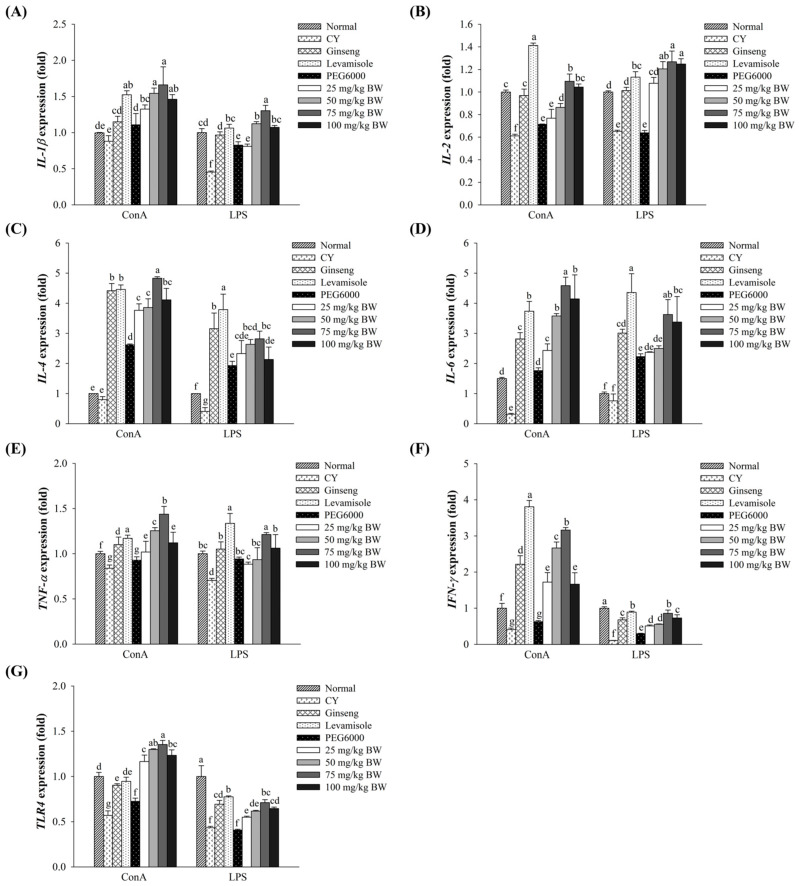
Effects of AJ-PEG on mRNA expression in splenic lymphocytes. Expression levels of (**A**) *IL-1β*, (**B**) *IL-2*, (**C**) *IL-4*, (**D**) *IL-6*, (**E**) *TNF-α,* (**F**) *IFN-γ*, and (**G**) *TLR4* are shown. All values are presented as means ± SD. Different letters indicate a statistically significant difference at *p* < 0.05 within treatment groups.

**Figure 4 foods-12-03292-f004:**
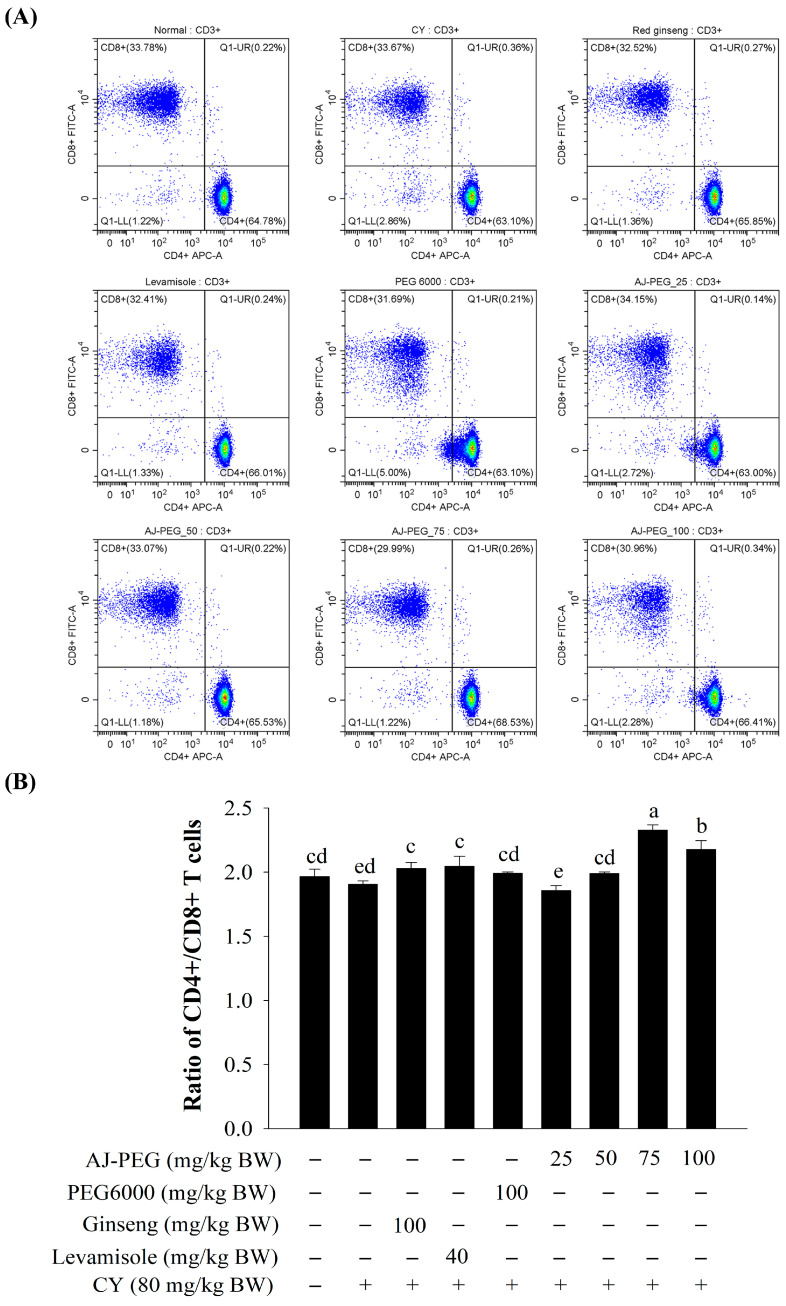
Effect of AJ-PEG on T-lymphocyte subsets of the spleen. (**A**) Representative flow cytometry analysis. (**B**) Percentages of CD4+ and CD8+ T cell subsets. Values are presented as mean ± SD. Different letters indicate a statistically significant difference at *p* < 0.05 within treatment groups.

**Figure 5 foods-12-03292-f005:**
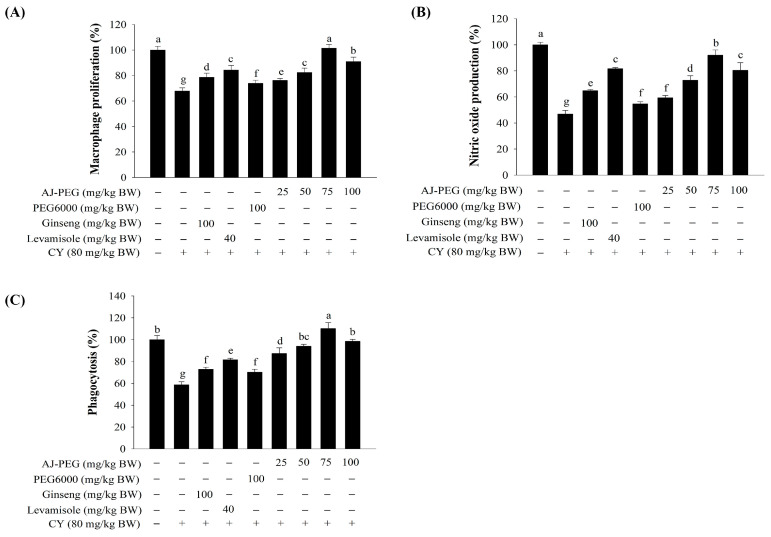
Effects of AJ-PEG on peritoneal macrophages. (**A**) Macrophage proliferation. (**B**) NO production. (**C**) Phagocytosis. Values are presented as means ± SD. Different letters indicate a statistically significant difference at *p* < 0.05 within treatment groups.

**Figure 6 foods-12-03292-f006:**
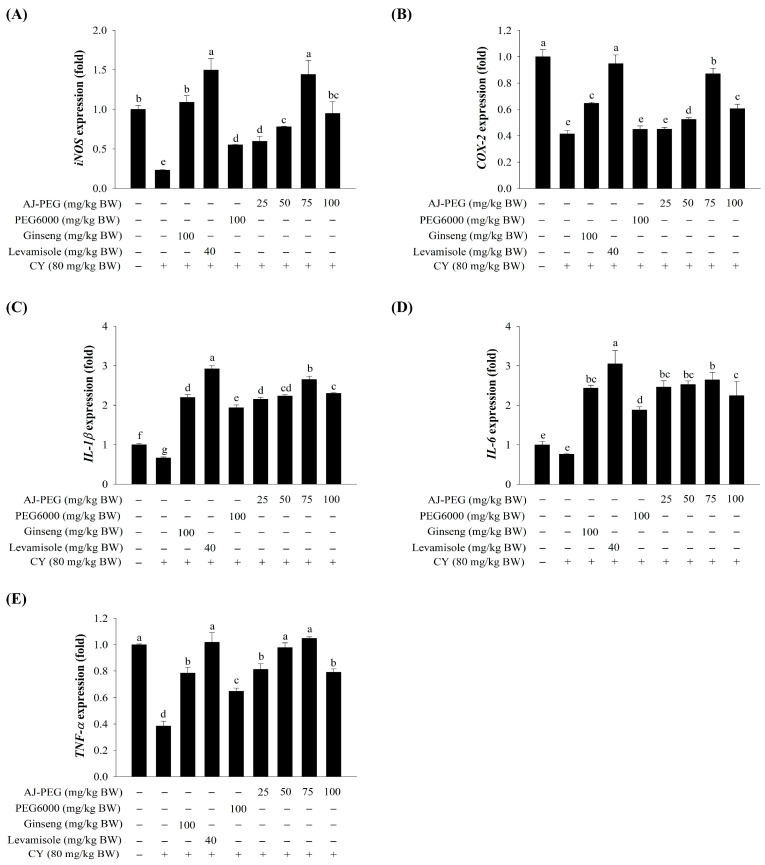
Effects of AJ-PEG on mRNA expression of immune genes in LPS-stimulated peritoneal macrophages. Expression levels of (**A**) *iNOS*, (**B**) *COX-2*, (**C**) *IL-1β*, (**D**) *IL-6*, and (**E**) *TNF-α* are shown. Values are presented as mean ± SD. Different letters indicate a statistically significant difference at *p* < 0.05 within treatment groups.

**Table 1 foods-12-03292-t001:** Mouse groups for immunosuppression induction and treatments.

Group	Dose (mg/kg BW)	Treatment		
		Day 1 to 3 (Orally)	Day 4 to 6 (Orally + i.p.)	Day 7 to 10 (Orally)
Normal	-	Saline	Saline	Saline
CY	80	Saline	Saline + CY	Saline
Ginseng	100	Ginseng	Ginseng + CY	Ginseng
Levamisole	40	Levamisole	Levamisole + CY	Levamisole
PEG6000	100	PEG6000	PEG6000+CY	PEG6000
AJ-PEG	25	AJ-PEG	AJ-PEG + CY	AJ-PEG
	50	AJ-PEG	AJ-PEG + CY	AJ-PEG
	75	AJ-PEG	AJ-PEG + CY	AJ-PEG
	100	AJ-PEG	AJ-PEG + CY	AJ-PEG

**Table 2 foods-12-03292-t002:** Sequences of gene-specific primers used for real-time PCR.

Gene	Accession No.	Sequence of Primer (5′ to 3′)	
		Forward Primer	Reverse Primer
iNOS	BC062378.1	TTCCAGAATCCCTGGACAAG	TGGTCAAACTCTTGGGGTTC
IL-1β	NM_008361.4	GGGCCTCAAAGGAAAGAATC	TACCAGTTGGGGAACTCTGC
IL-2	NM_008366.3	CCTGAGCAGGATGGAGAATTACA	TCCAGAACATGCCGCAGAG
IL-4	NM_021283.2	ACAGGAGAAGGGACGCCAT	GAAGCCCTACAGACGAGCTCA
IL-6	NM_031168.2	AGTTGCCTTCTTGGGACTGA	CAGAATTGCCATTGCACAAC
IFN-γ	NM_008337.3	CTCAAGTGGCATAGATGT	GAGATAATCTGGCTCTGCAGGATT
TNF-α	D84199.2	ATGAGCACAGAAAGCATGATC	TACAGGCTTGTCACTCGAATT
TLR4	NM_021297.3	CGCTCTGGCATCATCTTCAT	GTTGCCGTTTCTTGTTCTTCC
COX-2	NM_011198.4	AGAAGGAAATGGCTGCAGAA	GCTCGGCTTCCAGTATTGAG
β-actin	NM_007393.5	CCACAGCTGAGAGGGAAATC	AAGGAAGGCTGGAAAAGAGC

## Data Availability

All data of this study are presented in the manuscript.

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
