# Peer review of "Arctoscopus japonicus Lipids Enhance Immunity of Mice with Cyclophosphamide-Induced Immunosuppression"

_foods, 2023, doi:10.3390/foods12173292_

Round 1
Reviewer 1 Report
General comments:
When possible change lipid in AJ
In the Introduction: the capacity of fatty acids to obtain positive treatment of diseases should be limited. Maybe, serious disease could be avoided simply increasing the input of some selected PUFAs.
The relation between chronic diseases and immune regulation and oxidative stress should be carefully considered, highlighting better the possible utility and impact of the research on ordinary and common diseases, in particular in the Conclusions.
About the Discussion: many sentences were already reported in the MN, I suggest to limit the arguments to the novelties of the research.
In particular,
Lines 41-45 the sentence: Lipids and fatty acids are critical structural components of cell membranes and are essential for several metabolic processes as well as cellular functions [11]. Marine fish eggs contain high levels of polyunsaturated fatty acids (PUFAs) such as eicosapentaenoic acid (EPA; 20:5n3) and docosahexaenoic acid (DHA; 22:6n3), which provide energy for the fish to survive and reproduce [12] should be changed in: Lipids and fatty acids are main structural components of cell membranes and are essential for several metabolic processes, as well as some cellular functions [11]. Marine fish eggs contain high levels of polyunsaturated fatty acids (PUFAs), such as eicosapentaenoic acid (EPA; 20:5n3) and docosahexaenoic acid (DHA; 22:6n3), which provide energy for the fish to survive and reproduce [12].
Lines 47-49 I suggest to limit the emphasis of the sentence Researchers have shown that fatty acids reduce inflammatory responses such as Alzheimer’s disease, cancer disease, cardio-vascular disease, and anti-inflammatory conditions [13-16] changing in Researchers have shown that fatty acids may be useful to reduce inflammatory responses, such as Alzheimer’s disease, cancer, cardio-vascular disease, and anti-inflammatory conditions [13-16] and the same consideration may be valid for other sentences. Unfortunately, fatty acids are not the real solution for all these pathologies, which need much more complex therapeutic approaches.
Lines 59-60 Saturated fatty acids (SFAs) such as palmitic acid (16:0) and 59 stearic acid (18:0) have also been involved to immune response change in Saturated fatty acids (SFAs), such as palmitic acid (16:0) and
stearic acid (18:0), have also been involved to immune response
Line 70 clarify and reconsider the meaning of the sentence
Line 77 change the sentence: Six-week-old BALB/c mice with a weight of 21–23 g was provided by the Central Lab in: Six-week-old BALB/c mice with a weight of 21–23 g were provided by the Central Lab
Page 2 lines 86-87 Describe the procedure of the GC/MS analyses, including the utilized instrumentation and the use of standards or other identification methods, as well calibration.
Page 3 line 125 37°C change in 37 °C and check for other similar formal errors.
some sentences must be revised also to help the possibility of the reader to understand the meaning. Not always the reader is a specialist of the matter, but could be interested in the application of the research, also in other fields
Author Response
Lines 41-45 the sentence: Lipids and fatty acids are critical structural components of cell membranes and are essential for several metabolic processes as well as cellular functions [11]. Marine fish eggs contain high levels of polyunsaturated fatty acids (PUFAs) such as eicosapentaenoic acid (EPA; 20:5n3) and docosahexaenoic acid (DHA; 22:6n3), which provide energy for the fish to survive and reproduce [12] should be changed in: Lipids and fatty acids are main structural components of cell membranes and are essential for several metabolic processes, as well as some cellular functions [11]. Marine fish eggs contain high levels of polyunsaturated fatty acids (PUFAs), such as eicosapentaenoic acid (EPA; 20:5n3) and docosahexaenoic acid (DHA; 22:6n3), which provide energy for the fish to survive and reproduce [12].
- It was corrected.
- Thanks.
Lines 47-49 I suggest to limit the emphasis of the sentence Researchers have shown that fatty acids reduce inflammatory responses such as Alzheimer’s disease, cancer disease, cardio-vascular disease, and anti-inflammatory conditions [13-16] changing in Researchers have shown that fatty acids may be useful to reduce inflammatory responses, such as Alzheimer’s disease, cancer, cardio-vascular disease, and anti-inflammatory conditions [13-16] and the same consideration may be valid for other sentences. Unfortunately, fatty acids are not the real solution for all these pathologies, which need much more complex therapeutic approaches.
- As you suggested, it was corrected and the associated clinical references were also added.
- Skulas-Ray, A. C. (2015). Omega-3 fatty acids and inflammation: A perspective on the challenges of evaluating efficacy in clinical research. Prostaglandins & other lipid mediators, 116, 104-111.
- Flock, M. R., Skulas-Ray, A. C., Harris, W. S., Gaugler, T. L., Fleming, J. A., & Kris-Etherton, P. M. (2014). Effects of supplemental long-chain omega-3 fatty acids and erythrocyte membrane fatty acid content on circulating inflammatory markers in a randomized controlled trial of healthy adults. Prostaglandins, Leukotrienes and Essential Fatty Acids, 91(4), 161-168.
- Pischon, T., Hankinson, S. E., Hotamisligil, G. S., Rifai, N., Willett, W. C., & Rimm, E. B. (2003). Habitual dietary intake of n-3 and n-6 fatty acids in relation to inflammatory markers among US men and women. Circulation, 108(2), 155-160.
- Polyviou, T., MacDougall, K., Chambers, E. S., Viardot, A., Psichas, A., Jawaid, S., ... & Morrison, D. J. (2016). Randomised clinical study: inulin short‐chain fatty acid esters for targeted delivery of short‐chain fatty acids to the human colon. Alimentary pharmacology & therapeutics, 44(7), 662-672.
Lines 59-60 Saturated fatty acids (SFAs) such as palmitic acid (16:0) and 59 stearic acid (18:0) have also been involved to immune response change in Saturated fatty acids (SFAs), such as palmitic acid (16:0) and stearic acid (18:0), have also been involved to immune response
- It was corrected.
- Thanks
Line 70 clarify and reconsider the meaning of the sentence
- It was corrected.
- Thanks
Line 77 change the sentence: Six-week-old BALB/c mice with a weight of 21–23 g was provided by the Central Lab in: Six-week-old BALB/c mice with a weight of 21–23 g were provided by the Central Lab
- It was corrected.
- Thanks
Page 2 lines 86-87 Describe the procedure of the GC/MS analyses, including the utilized instrumentation and the use of standards or other identification methods, as well calibration.
Page 3 line 125 37°C change in 37 °C and check for other similar formal errors.
- It was corrected.
- Thanks.
Comments on the Quality of English Language
some sentences must be revised also to help the possibility of the reader to understand the meaning. Not always the reader is a specialist of the matter, but could be interested in the application of the research, also in other fields
- It was corrected.
- Thanks

Reviewer 2 Report
In this study, the authors extracted lipid from the eggs of Arctoscopus japonicus, and incorporated with PEG6000 to produce AJ-PEG, and its immunomodulatory function was investigated on the CY-induced immunosuppressed mice. This study was well designed and the results were promising. Therefore, in my opinion, this manuscript is suitable for the publication in this journal after finishing the following minor revisions:
1. In the title, it will be clearer if “Cyclophosphamide-induced Mice” can be revised to “Cyclophosphamide-induced Immunosuppressed Mice”
2. Line 33-34: They exert immune organ indices …? This sentence needs to be revised.
3. Line 90: The authors should briefly explain the purpose of using PEG6000.
4. Line 11, 32, 95: Immunosuppressed mice, immunosuppressive mice and Immunosuppression Mouse. The authors should use a uniform description.
5. Line 96: What is the meaning of “After adaptive breeding”? Is it “After environmental adaption”?
6. Line 168: “There have been several studies that have shown…” This sentence needs to be revised.
7. Line 271-275, the introduction of CY is not necessary.
8. Line 292, AJ-PER?
Minor editing of English language required
Author Response
In this study, the authors extracted lipid from the eggs of Arctoscopus japonicus, and incorporated with PEG6000 to produce AJ-PEG, and its immunomodulatory function was investigated on the CY-induced immunosuppressed mice. This study was well designed and the results were promising. Therefore, in my opinion, this manuscript is suitable for the publication in this journal after finishing the following minor revisions:
- In the title, it will be clearer if “Cyclophosphamide-induced Mice” can be revised to “Cyclophosphamide-induced Immunosuppressed Mice”
- It was corrected.
- Thanks.
- Line 33-34: They exert immune organ indices …? This sentence needs to be revised.
- It was corrected.
- Thanks.
- Line 90: The authors should briefly explain the purpose of using PEG6000.
- It was corrected in lines 96-99 and 290-294.
- Thanks.
- Line 11, 32, 95: Immunosuppressed mice, immunosuppressive mice and Immunosuppression Mouse. The authors should use a uniform description.
- It was corrected.
- Thanks.
- Line 96: What is the meaning of “After adaptive breeding”? Is it “After environmental adaption”?
- It was corrected.
- Thanks.
- Line 168: “There have been several studies that have shown…” This sentence needs to be revised.
- It was corrected.
- Thanks.
- Line 271-275, the introduction of CY is not necessary.
- It was corrected.
- Thanks.
- Line 292, AJ-PER?
- It was corrected.
- Thanks,

Reviewer 3 Report
This manuscript reports the immunostimulatory effects of lipids extracted of Arctoscopus japonicus eggs incorporated with PEG6000 in immunosuppressed mice treated with cyclophosphamide (CY). The aims of the manuscript are interesting and the work was pparently well conducted. However, there are some points that need explanation and/or correction. Please see the comments below.
1- Abstract:
- Line 10: “The lipid extract was extracted from...” – please rephrase to “The lipid extract was obtained from...”.
- Line 15: “The proliferation of splenic lymphocytes and NK activity were also stimulated...” – please write NK in full as in cyclophosphamide (CY);
- Line 19: “...NO production, and phagocytosis...” - please write NO full as in cyclophosphamide (CY);
2- Materials and Methods:
- Please initially provide a subsection on the chemicals used in the different experimental steps, containing information about the manufacturer, city and country in parentheses;
- Line 77: “Six-week-old BALB/c mice with a weight of 21–23 g was provided ...” – please rephrase to: “Six-week-old BALB/c mices with a weight of 21–23 g were provided...”;
- Line 83: “A. japonicus was discovered in Gangwon...” – please rephrase to “A. japonicus was obtained in Gangwon ...”;
- Lines 85 to 87: “The gas chromatography (GC) -flame ionization detection (FID) analysis was used to determine the fatty acid profiles of A. japonicus lipids” – please provide a complete description of the equipment and analysis conditions employed;
- Line 88: “absolute alcohol”. – please rephrase to “absolute ethanol”;
- Line 118: 2.7 Determination of Peritoneal Macrophage Proliferation and NO Production – please write NO in full;
- Line 119: “LPS...” - please write abbreviation in full the first time it is mentioned in the text and review the entire manuscript;
- Line 124: “WST...” - please write in full;
- Line 133: Con A...” - please write in full;
3- Results
- The quality of the Figures needs to be improved. Please, do not use gray color in the axes and legends, but black color. Also increase the size of the fonts used in the axes and legends within the figures, especially in figures 2, 3, 4B, 5 and 6. Also, Figure 4A is unreadable and needs to be completely improved.
- Please present Figure 5 immediately after mentioning it in the text, that is, before item 3.8 (Effect of AJ-PEG on Peritoneal Macrophage Phagocytosis);
4- Discussion
- Line 269: “macrophages, in vitro cellular system...” – please rephrase to “macrophages, in in vitro cellular system...”;
Minor editing of English language required.
Author Response
This manuscript reports the immunostimulatory effects of lipids extracted of Arctoscopus japonicus eggs incorporated with PEG6000 in immunosuppressed mice treated with cyclophosphamide (CY). The aims of the manuscript are interesting and the work was pparently well conducted. However, there are some points that need explanation and/or correction. Please see the comments below.
1- Abstract:
- Line 10: “The lipid extract was extracted from...” – please rephrase to “The lipid extract was obtained from...”.
- It was corrected.
- Thanks.
- Line 15: “The proliferation of splenic lymphocytes and NK activity were also stimulated...” – please write NK in full as in cyclophosphamide (CY);
- It was corrected.
- Thanks.
- Line 19: “...NO production, and phagocytosis...” - please write NO full as in cyclophosphamide (CY);
- It was corrected.
- Thanks.
2- Materials and Methods:
- Please initially provide a subsection on the chemicals used in the different experimental steps, containing information about the manufacturer, city and country in parentheses;
- Line 77: “Six-week-old BALB/c mice with a weight of 21–23 g was provided ...” – please rephrase to: “Six-week-old BALB/c mices with a weight of 21–23 g were provided...”;
- It was corrected.
- Thanks.
- Line 83: “A. japonicus was discovered in Gangwon...” – please rephrase to “A. japonicus was obtained in Gangwon ...”;
- It was corrected.
- Thanks.
- Lines 85 to 87: “The gas chromatography (GC) -flame ionization detection (FID) analysis was used to determine the fatty acid profiles of A. japonicus lipids” – please provide a complete description of the equipment and analysis conditions employed;
- Line 88: “absolute alcohol”. – please rephrase to “absolute ethanol”;
- It was corrected.
- Thanks.
- Line 118: 2.7 Determination of Peritoneal Macrophage Proliferation and NO Production – please write NO in full;
- It was corrected.
- Thanks.
- Line 119: “LPS...” - please write abbreviation in full the first time it is mentioned in the text and review the entire manuscript;
- It was corrected.
- Thanks.
- Line 124: “WST...” - please write in full;
- It was corrected.
- Thanks.
- Line 133: Con A...” - please write in full;
- It was corrected.
- Thanks.
3- Results
- The quality of the Figures needs to be improved. Please, do not use gray color in the axes and legends, but black color. Also increase the size of the fonts used in the axes and legends within the figures, especially in figures 2, 3, 4B, 5 and 6. Also, Figure 4A is unreadable and needs to be completely improved.
- It was corrected.
- Thanks.
- Please present Figure 5 immediately after mentioning it in the text, that is, before item 3.8 (Effect of AJ-PEG on Peritoneal Macrophage Phagocytosis);
- It was corrected.
- Thanks.
4- Discussion
- Line 269: “macrophages, in vitro cellular system...” – please rephrase to “macrophages, in in vitro cellular system...”;
- It was corrected.
- Thanks.

Round 2
Reviewer 1 Report
Changes are in accordance with the comments. The paper can be published.